# Leflunomide Treatment Does Not Protect Neural Cells following Oxygen-Glucose Deprivation (OGD) In Vitro

**DOI:** 10.3390/cells13070631

**Published:** 2024-04-04

**Authors:** Claire J. M. Curel, Irene Nobeli, Claire Thornton

**Affiliations:** 1Department of Comparative Biomedical Sciences, Royal Veterinary College, London NW1 0TU, UK; 2School of Natural Sciences, Institute of Structural and Molecular Biology, Birkbeck, University of London, London WC1E 7HX, UK

**Keywords:** Leflunomide, neonatal, hypoxia-ischemia, oxygen-glucose deprivation, mitochondria, OPA1, mitofusins

## Abstract

Neonatal hypoxia-ischemia (HI) affects 2–3 per 1000 live births in developed countries and up to 26 per 1000 live births in developing countries. It is estimated that of the 750,000 infants experiencing a hypoxic-ischemic event during birth per year, more than 400,000 will be severely affected. As treatment options are limited, rapidly identifying new therapeutic avenues is critical, and repurposing drugs already in clinical use offers a fast-track route to clinic. One emerging avenue for therapeutic intervention in neonatal HI is to target mitochondrial dysfunction, which occurs early in the development of brain injury. Mitochondrial dynamics are particularly affected, with mitochondrial fragmentation occurring at the expense of the pro-fusion protein Optic Atrophy (OPA)1. OPA1, together with mitofusins (MFN)1/2, are required for membrane fusion, and therefore, protecting their function may also safeguard mitochondrial dynamics. Leflunomide, an FDA-approved immunosuppressant, was recently identified as an activator of MFN2 with partial effects on OPA1 expression. We, therefore, treated C17.2 cells with Leflunomide before or after oxygen-glucose deprivation, an in vitro mimic of HI, to determine its efficacy as a neuroprotection and inhibitor of mitochondrial dysfunction. Leflunomide increased baseline OPA1 but not MFN2 expression in C17.2 cells. However, Leflunomide was unable to promote cell survival following OGD. Equally, there was no obvious effect on mitochondrial morphology or bioenergetics. These data align with studies suggesting that the tissue and mitochondrial protein profile of the target cell/tissue are critical for taking advantage of the therapeutic actions of Leflunomide.

## 1. Introduction

Birth asphyxia deprives the neonatal brain of blood flow, impairing nutrient supply to the brain around the time of birth and leading to hypoxic-ischemic encephalopathy (HIE). Moderate to severe HIE has an incidence of between 2 and 3 per 1000 live births in the UK [1], but these rates rise significantly in low-resource settings [2,3]. The consequences are life-long, ranging from mild impairments to significant motor, cognitive, or sensory disabilities such as cerebral palsy or epilepsy, with a severe impact on the quality of life of the patient and their families [4]. The primary phase of the injury during the asphyxial event is characterized by a significant depletion of high-energy phosphates driven by the lack of substrate. This is followed by the latent phase, where brain energy levels are briefly restored and can act as a window for intervention. The secondary phase is represented by a prolonged energy failure, and it is during this time that the majority of predominantly neuronal cell death occurs and longer-term cellular pathologies are triggered e.g., inflammation, astrogliosis [5]. The available treatment for this injury is therapeutic hypothermia instigated within the latent phase, but although it provides proof-of-concept that intervention following injury can provide neuroprotection, it is fully successful for only 1 in 7 babies [6]. Additionally, there is debate over its use and efficacy in low-resource settings where arguably an HIE therapy is most urgently needed [7]. 

Key characteristics of this injury, e.g., impaired ATP production, increased intracellular Ca^2+^, and increased ROS production, implicate mitochondrial dysfunction as a central player in the neurocellular response to birth asphyxia. Increased calcium buffering results in mitochondrial swelling and cristae rupture [8], releasing proapoptotic molecules such as cytochrome c into the mitochondrial intermembrane space. These proapoptotic molecules are then leaked into the cytosol via heterodimeric Bax/Bak pores on the outer mitochondrial membrane. Once in the cytosol, cytochrome c can form apoptosomes by combining with Apaf-1 and procaspase 9. This then activates caspase 9 and caspase 3, causing chromatin condensation, cytoplasmic bleb, and apoptotic body formation, leading to apoptotic cell death [9]. Increased Ca^2+^ and NO influx, as well as the escape of electrons from mitochondrial respiratory complexes, result in increased ROS production and damage to cell membranes [10,11]. 

Mitochondrial dynamics, or the fission and fusion of mitochondria to maintain function, are emerging as targets for neonatal HI. Mitochondrial fission is governed by the action of Dynamic related protein (DRP)1, whereas mitochondrial fusion is carried out by mitofusins (MFN1/2) and Optic Atrophy (OPA)1 fusing the outer and inner mitochondrial membranes, respectively [12]. In a well-characterized rodent model of neonatal HI, rapid mitochondrial fragmentation was observed following injury, followed by an increase in mitophagy (mitochondrial recycling), followed by a second wave of mitophagy several days later [13]. Using the same model, we and others have shown that OPA1 is itself cleaved into shorter, potentially fusion-incompetent forms [14,15]. Inhibiting mitochondrial fission or, conversely, enhancing mitochondrial fusion may, therefore, protect mitochondrial integrity and cell health.

Considerable effort has been made to intervene at points in the apoptotic pathway triggered by mitochondrial dysfunction in the neonatal brain. For example, inhibition of Bax/Bak themselves [16,17], targeting of initiator caspases [18,19,20], or modulation of apoptosis-inducing factor [21,22] have all shown promise in reducing infarct size and improving behavioral outcomes in preclinical models of neonatal HI. However, these have not been successfully translated to the clinic so far. Repurposing drugs already in use for other clinical indications may offer a more rapid route for therapy development and may also increase the likelihood of translational success [23]. A recent screening study identified Leflunomide, an FDA-approved drug used for rheumatoid arthritis treatment, as an activator of mitochondrial fusion, increasing the expression of mitofusins in human HeLa cells and mouse embryonic fibroblasts and muscle cell lines [24]. The beneficial effects of Leflunomide on mitochondrial fusion were also demonstrated in a human breast cancer cell line [25]. Previous studies have also tested the effects of Leflunomide treatment before ischemia-reperfusion injury. The antioxidant and anti-inflammatory effects of Leflunomide were shown to rescue ischemic tissue damage in the intestine [26], liver [27], and kidney [28] in adult rats. We aimed to test this drug on a neural cell line to determine if Leflunomide treatment could promote mitochondrial fusion and, therefore, enhance cell survival following oxygen-glucose deprivation, an in vitro mimic of HI. 

## 2. Materials and Methods

### 2.1. Cell Culture

The mouse cerebellar neural progenitor stem-like cell line C17.2 (ECACC 07062902, Sigma-Aldrich, Poole, UK [29]) and human DSRed-HeLa cells (a kind gift from Prof Haya Lorberboum-Galski, HUJI [30,31]) were cultured in heat-inactivated Dulbecco’s Modified Eagle’s Medium (DMEM) supplemented 10% fetal bovine serum (FBS), 1% penicillin/streptomycin, 25 mM glucose and 2 mM L-Glutamine at 37 °C, 5% CO_2_ in a humidified incubator. Cells were passed twice weekly.

### 2.2. Oxygen-Glucose Deprivation (OGD)

C17.2 cells were subjected to oxygen-glucose deprivation, as described previously [14]. Briefly, growth medium was replaced with degassed artificial cerebrospinal fluid (aCSF; 300 mM NaCl, 6 mM KCl, 2.8 mM CaCl_2_, 1.6 mM MgCl_2_, 1.6 mM Na_2_ HPO_4_, 0.4 mM NaH_2_PO_4_) and incubated at 37 °C in 95%N_2_/5%CO_2_. Following the incubation, aCSF was removed and replaced with fresh growth medium, and the cells were returned to normoxia at 37 °C, which is 5% CO_2_. The control plate medium was changed at the beginning of the incubation period and again after 3 h of OGD.

### 2.3. Nuclear and Mitochondrial Staining in Live Cells

Hoechst 33342 (10 μg/mL final concentration) and/or Mitotracker Orange (100 nM, ThermoFisher) were added to cells in growth medium and plates incubated in the dark (37 °C, 5% CO_2_). Following a 30-min incubation, the medium was aspirated, the cells washed in sterile PBS, and the fresh growth medium returned. Stained cells were imaged using an EVOS M5000 microscope and counted using the SparkCyto-400 plate reader (TECAN, Männedorf, Switzerland). Mitotracker orange staining was further visualized using the Leica DM4000 upright fluorescence microscope (Leica, Wetzlar, Germany).

### 2.4. Leflunomide Treatment

The pre-OGD and post-OGD treatment timeline for Leflunomide (CAY14860, Cayman Chemical) is shown in Figure 1. Leflunomide stock (100 mM stock) was diluted in a medium and added to cells in varying concentrations (0–200 μM). Cells were incubated for 16 h and processed according to the timeline.

### 2.5. Mitochondrial Stress Test Assay (Seahorse)

Cells were plated on 96-well Seahorse plates (Agilent, Stockport, UK) at a concentration of 5000 cells per well and treated with Leflunomide (25, 50 μM) for 16 h. Seahorse cartridges were prepared according to the manufacturer’s instructions. DMEM assay medium (Agilent) was supplemented with 1 M glucose, 100 mM pyruvate, and 200 mM glutamine and used to prepare the mitochondrial assay compounds (Table 1). Once the cartridge was loaded with the appropriate compounds, the assay cartridge and cell plates were loaded into a Seahorse Xf Pro bioanalyzer and run with standard parameters. Following the assay, cells were stained with Hoechst, and assay data were normalized to the cell counts for each well.

### 2.6. Western Blot

C17.2 and HeLa cells were harvested in Hepes Buffer A (50 mM HEPES (pH 7.5)), 50 mM sodium fluoride, 5 mM sodium pyrophosphate, 1 mM EDTA, and protease inhibitors (Merck) containing 1% Triton X-100, incubated on ice for 20 min and sonicated. The lysate was centrifuged to remove insoluble material, and the protein concentration was determined using Bradford assay according to the manufacturer’s instructions (Biorad, Watford, UK). Absorbance was measured at 590 nm, and protein concentration was interpolated from a bovine serum albumin standard curve. 

Protein lysates (50 μg) were resolved on 4–12% Bis-Tris SDS-page gels in NuPage MOPS buffer (Thermo Fisher Scientific, Loughborough, UK) and transferred to polyvinylidene fluoride membrane PVDF membrane (Immobilon FL, Merck, Gillingham, UK) specialized for low fluorescent background. Western blot and LI-COR fluorescence imaging were carried out at 680/800 nm using an Odyssey DLx imager (LI-COR Biosciences, Cambridge, UK) as described previously [14]. The following antibodies were used in this study: mouse anti-OPA1 (Clone 18/BD Bioscience 612607), mouse anti-MFN2 ([6A8], Abcam, Cambridge, UK), anti-mouse GAPDH (Merck G8795), IRDye 680RD and 800CW goat anti-mouse secondary antibodies (LI-COR Biosciences). Densitometric quantification of OPA1, MFN2, and GAPDH was performed using Image Studio software v5.2 (LI-COR Biosciences). 

### 2.7. Image J Analyses

Mitochondria images taken using the DM4000 upright fluorescence microscope were analyzed using the MINA plugin in Image J [32]. The parameters used for analysis were ‘individual’, referring to the number of mitochondria with 0 or 1 branch, ‘mitochondrial footprint’, referring to the area of the cell incorporating mitochondria, ‘network’, referring to the number of branched mitochondria, and ‘mean network size’ which refers to the mean number of branches per mitochondria in each image. 

### 2.8. Statistical Analysis

Statistical analysis was conducted using Prism Software (v9, GraphPad). Data were evaluated for normality using the Shapiro–Wilk test and then assessed using one-way or two-way ANOVA, followed by *post hoc* tests as appropriate, or for linear trend. Details of tests used are highlighted in the text, and a *p*-value of 0.05 or less was deemed significant. Biological replicates for cell line analyses are defined as experiments on different passages of cells, and each biological replicate comprises a minimum of two technical replicates. 

## 3. Results

### 3.1. Identification of the Optimal Leflunomide Concentration for C17.2 Cells

Enhanced mitochondrial fusion following cellular stress may provide cells with a means to prevent the induction of cell death pathways [33]. Leflunomide is reported to induce mitochondrial fusion [24], and preclinical models of neonatal brain injury have observed induction of mitochondrial fission; therefore, Leflunomide may have the potential for drug repurposing for this indication. We began our evaluation in vitro, examining whether Leflunomide could provide neuroprotection in a neural stem/precursor cell line, C17.2, a line derived from neonatal cerebellum [34] and which has the capability of differentiating into neurons and astrocytes [29]. Both pro-survival and pro-death properties have been ascribed to Leflunomide [26,35]; therefore, we measured the effect of increasing concentrations on C17.2 cell number 24 h post-addition. Testing for linear trend revealed that there was a significant negative slope (−0.1588, *p* = 0.0043), and one-way ANOVA identified an overall significant difference (*p* = 0.0465) most noticeable at the higher concentrations (* *p* = 0.0142, Figure 2). As 50 µM Leflunomide did not cause significant toxicity compared with untreated control cells (*p* = 0.3270) and as this was a high enough dose to induce mitochondrial fusion as reported previously [24], we continued our experiments using this concentration. 

### 3.2. Leflunomide Treatment Does Not Alter Cell Survival following OGD

To evaluate whether Leflunomide could promote neuroprotection in immature neural cells following neonatal HI injury, we used oxygen/glucose deprivation (OGD), an in vitro mimic of HI. C17.2 cells were pretreated with Leflunomide (50 μM) for 16 h and then subjected to OGD (3 h). Cells were “reperfused” with a growth medium for 1 h after the injury, and cell number was determined. In control untreated cells, OGD caused 50% cell death (Figure 3a, *p* = 0.0151, two-way ANOVA); however, Leflunomide pretreatment could not prevent cell death after OGD (Figure 3a, Ctrl+Lef vs. OGD+Lef: *p* = 0.1689).

Clinical cases of birth asphyxia cannot be predicted, and therefore, any treatments must be efficacious when given post-insult. We, therefore, investigated whether Leflunomide could protect against OGD-mediated C17.2 cell death when administered to cells immediately after the insult. C17.2 cells were exposed to OGD for 3 h and then received Leflunomide. At 16 h post-insult, cell number was determined as previously. As expected, OGD caused a significant decrease in cell number (*p* = 0.0134, two-way ANOVA, Figure 3b); similarly to the pretreatment data in Figure 3a, post-insult treatment with Leflunomide did not rescue cells from OGD-mediated cell death (Figure 3b).

### 3.3. Leflunomide Treatment in C17.2 Cells Does Not Alter Mitochondrial Morphology 

To establish the effect of Leflunomide on mitochondria, we pretreated C17.2 cells (16 h) before staining them with mitotracker orange. Cells were imaged (Figure 4a), and mitochondrial morphology was quantified using the MiNA plugin in Image J (Figure 4b) [32]. OGD exposure resulted in a decrease in the complexity of mitochondrial networks compared with control (OGD, *p* = 0.0337), as shown by the decreases in branches per network, but Leflunomide could not rescue this impairment (OGD+Lef, *p* = 0.0462). 

### 3.4. Effect of Leflunomide on Mitochondrial Fusion Proteins

To investigate this lack of effect, we evaluated the expression of mitochondrial fusion proteins in response to Leflunomide treatment. As a positive control, we confirmed the previous observation that Leflunomide upregulates MFN2 protein expression in HeLa cells (Figure 5a [24]). However, this response was blunted for MFN2 in C17.2 cells (*p* = 0.1851, two-way ANOVA) regardless of whether the cells were exposed to OGD (Figure 5b,c). We also examined OPA1, the GTPase responsible for inner mitochondrial membrane fusion. Surprisingly, OPA1 expression increased after Leflunomide treatment in C17.2 cells; however, Leflunomide did not prevent the loss of OPA1 expression as a result of OGD, nor protect the L-OPA1:S-OPA1 ratio (Figure 5b,d,e). 

### 3.5. Leflunomide Pretreatment Increases Mitochondrial Proton Leak

In addition to promoting mitochondrial fusion, Leflunomide has been tested for its effects on mitochondrial bioenergetics, although there is currently no consensus on the outcome [24,36,37]. We, therefore, performed a mitochondrial stress test (Figure 6a) to determine whether pretreatment of C17.2 cells with Leflunomide (25&50 μM) impacted mitochondrial metrics. As with Miret-Casals et al. [24], we found that Leflunomide treatment did not affect basal or maximal mitochondrial respiration. However, there was a significant increase in proton leak at the higher Leflunomide concentration (Figure 6b).

## 4. Discussion

Hypoxic-ischemic injury during birth affects up to 750,000 babies per year and contributes 2.4% of the global burden of disease [38]. Therapeutic hypothermia is the only treatment currently available for these infants and can reduce the risk of disability and mortality. However, therapeutic hypothermia only has a beneficial effect in approximately 1 in every 7 babies treated [39], and it is currently unclear why this lack of efficacy occurs. Therapeutic hypothermia is not always available in low-resource countries and is not possible to use in preterm infants [40]. For these reasons, adjunct therapies that could improve the effectiveness of therapeutic hypothermia or even replace this therapy are needed. 

In this study, we have investigated the potential of repurposing the rheumatoid arthritis drug Leflunomide for neonatal brain injury. 

Efficient mitochondrial dynamics, i.e., the balance of fusion and fission, is required for mitochondrial respiration and is emerging as a common target in numerous pathologies [12], including neonatal hypoxic-ischemic encephalopathy [13,14,41]. As such, interventions are sought that promote mitochondrial fusion or inhibit fission and represent a new avenue for therapeutic development [42,43]. High throughput screening of in-use therapeutics (with known bioavailability and safety profiles) provides a faster route to clinic, and using this approach, Miret-Casals and colleagues recently identified Leflunomide as a regulator of mitofusins in human and mouse cell lines [24]. We have trialed Leflunomide on an immature neural cell line in an established in vitro model of ischemic injury, OGD, to examine its neuroprotective and mitochondrial properties. We found that Leflunomide was unable to protect C17.2 cells from OGD regardless of whether the treatment was given as a pretreatment or post-insult. In addition, although we were able to replicate the elevation of MFN2 in HeLa cells [24], there was no apparent upregulation of MFN2 in C17.2 cells, although we did find the Leflunomide increased total OPA1 protein expression in uninjured cells. However, this increase was abrogated by exposure to OGD, where OPA1 protein expression decreased regardless of the presence of Leflunomide. As we have previously observed in vivo, OGD reduced the ratio of L-OPA:S-OPA, indicating an increase in mitochondrial fragmentation [14]; Leflunomide was also unable to rectify this imbalance. Future work will continue to focus on how to protect OPA1 both from an overall reduction in expression and cleavage into shorter, potentially fusion-incompetent forms. 

Unfortunately, MFN2 levels were not significantly affected by Leflunomide treatment in C17.2 cells. Mitofusins are large GTPases needed to act in concert with OPA1 to enable the fusion of mitochondrial membranes. Agonists of MFN2 are reported to prevent pathological mitochondrial fragmentation and loss of mitochondrial membrane potential [44] and are therefore sought-after for several conditions exhibiting mitochondrial dysfunction. Such fragmentation is also apparent in mouse and cell models of neonatal HI [41,45], hence the testing of Leflunomide in our model systems. However, it is emerging that Leflunomide can have cell- and injury-specific effects, suggesting that results from other studies cannot be generally extrapolated. The Zorzano lab showed that Leflunomide promoted MFN2 expression in uninjured HeLa, C2C12 muscle cells, and mouse embryonic fibroblasts (MEFs, [24]). In contrast, Leflunomide-mediated increases in MFN2 were only apparent in hepatocarcinoma liver samples and absent in control liver samples; Leflunomide restored MFN2 to baseline after a cancer-induced decrease [46]. Leflunomide was also shown to restore MFN2 levels in human ovarian cancer cells [47]. This indicates that baseline expression of MFN2 may dictate whether Leflunomide acts on MFN2 levels. This is supported by findings in chronic obstructive pulmonary disease studies where Leflunomide recovered MFN2 expression previously reduced in lung tissues and alveolar cells exposed to cigarette smoke extract (CSE [48]). No effect of Leflunomide on MFN2 or OPA1 was observed in control cells. This phenomenon may explain why Leflunomide did not act on MFN2 levels in C17.2 cells, as we observed no substantial change in MFN2 levels following OGD. Miret-Casals et al. (2018) demonstrated that pyrimidine synthesis and MFN2 expression were linked by the ability of Leflunomide to inhibit pyrimidine synthesis via the inhibition of dihydroorotate dehydrogenase (DHODH). The subsequent depletion of the pyrimidine pool increased mitofusin expression and mitochondrial elongation [24]. It is likely that the routes to pyrimidine synthesis in different cell types (e.g., metabolically (over)active, resting, differentiated) may contribute to the tissue-specific effects of Leflunomide [49].

It is unlikely that the lack of Leflunomide MFN2 activity in our experiments was due to low-dose effects. Our initial dose-response assay suggested that concentrations higher than 50 μM would impact C17.2 cell survival, and this dose was originally used by Miret-Casals and colleagues [24]. Leflunomide (100 μM and above) has been reported as a tumor suppressor preventing cell proliferation and inducing apoptosis in a variety of cell and in vivo models [50,51], including neuroblastoma [35]. In our study, we identified a reduction in cell number, indicating cell death rather than a cytostatic effect on the cell cycle, preventing us from using higher concentrations.

Surprisingly, we identified that basal OPA1 expression increased in C17.2 after Leflunomide exposure, but this did not persist after OGD. This effect on OPA1 was reported to be cell-specific by Miret-Casals and colleagues, as they observed the phenomenon in MEFs but not HeLa and C2C12 cells. As with MFN, Leflunomide action on OPA1 may also vary with injury. In BEAS2B cells exposed to CSE, OPA1 levels were substantially reduced but restored following Leflunomide or BGP-15 treatment [52]. However, OGD resulted in a significant decrease in OPA1 expression that was not altered by Leflunomide. Finally, we observed an increase in mitochondrial proton leak following Leflunomide treatment (50 μM). Taken together with the impact of OGD on OPA1 expression and the loss of mitochondrial complexity, the addition of proton leak provoked by Leflunomide alone may explain why Leflunomide does not improve mitochondrial stability following OGD. It may be that this action of Leflunomide contributes to the induction of mitophagy, as observed by Yu and colleagues, when using the drug in their adenocarcinoma model [36]. 

The clinical unmet need for adjunct therapies for neonatal HIE has prompted several studies aiming to reposition already in-use treatments [38]. Potential repurposed therapies currently being prepared for, or in the process of, clinical trials for neonatal HIE include melatonin, allopurinol [53], and exendin-4 [38,54]. One of the most promising to date has been through the use of erythropoietin (EPO) [55]. EPO has known neuroprotective effects and showed promising results in a stage II clinical trial, where infants treated with both EPO and therapeutic hypothermia presented with lesser MRI-assessed brain injury and improved motor function in the short term [56]. However, a stage III clinical trial testing the efficacy of a high dose regimen found that combined EPO and hypothermia did not decrease the risk of death, nor did it improve motor impairments compared with therapeutic hypothermia alone, and potentially increased the risk of serious adverse events such as thrombosis [57,58]. However, as EPO and therapeutic hypothermia may have similar mechanisms of action, EPO may offer potential as a stand-alone therapy that would benefit low-resource settings where therapeutic hypothermia is unavailable [54].

## 5. Conclusions

In summary, we have tested Leflunomide for its neuroprotective characteristics in promoting mitochondrial fusion following OGD. Leflunomide increased OPA1 rather than MFN2 expression in control C17.2 cells but was unable to prevent mitochondrial impairment and subsequent cell death following OGD. Taken together with other studies, it is clear that the cellular context and mitochondrial protein profile of the target cell/tissue are critical for the therapeutic actions of Leflunomide on mitochondrial fusion. 

## Figures and Tables

**Figure 1 cells-13-00631-f001:**
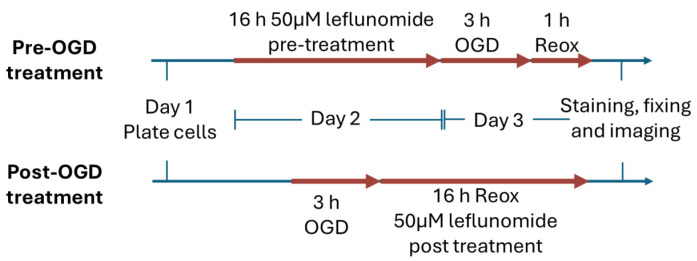
Timeline of Leflunomide experiments. Leflunomide was used as a pretreatment (upper line) or post-treatment following oxygen-glucose deprivation (OGD) in C17.2 cells. Reox: reoxygenation.

**Figure 2 cells-13-00631-f002:**
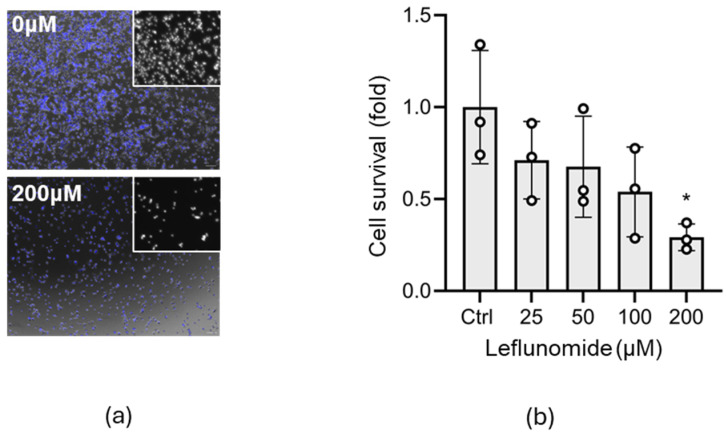
High concentrations of Leflunomide are toxic to C17.2 cells. C17.2 cells were treated with varying concentrations of Leflunomide (0–200 µM) for 24 h. (**a**) Cell nuclei were stained with Hoechst (blue), imaged, and cell number calculated. Inset shows nuclei alone. Scale bar = 25 µm (**b**) The graph shows fold change in cell number compared with untreated control. Data are expressed as mean ± SD and were analyzed by one-way ANOVA followed by Dunnett’s post hoc test (N = 3, with 3 technical repeats, * *p* < 0.05).

**Figure 3 cells-13-00631-f003:**
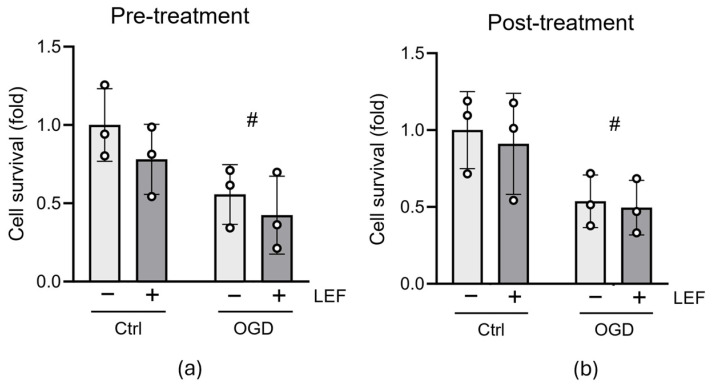
Leflunomide does not alter cell survival when administered either before or after OGD. (**a**) C17.2 cells were treated with Leflunomide (50 μM) for 16 h and then subjected to OGD (3 h). Following a subsequent 1 h recovery period, cells were treated with Hoechst, imaged, and quantified. (**b**) C17.2 cells were exposed to 3 h OGD, followed by Leflunomide treatment (50 μM). After 16 h, the cell number was quantified as above. Both graphs depict fold change in cell number compared with untreated control. Data are expressed as mean ± SD and were analyzed by two-way ANOVA followed by Tukey post hoc test (N = 3, with 3 technical repeats, # *p* < 0.05 for injury).

**Figure 4 cells-13-00631-f004:**
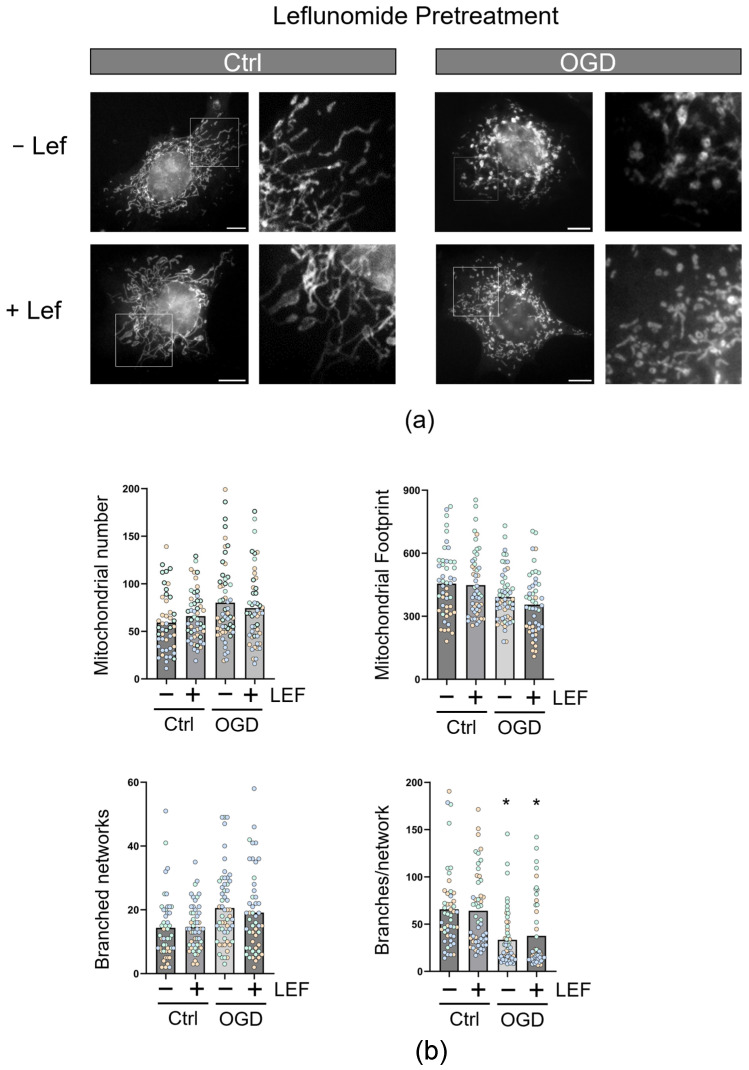
Leflunomide pretreatment does not alter mitochondrial morphology in C17.2, regardless of injury. (**a**) C17.2 cells were treated with Leflunomide (50 μM) for 16 h and/or OGD (3 h). Following a subsequent 1 h recovery period, cells were treated with mitotracker orange and imaged. Scale bar represents 2 μm. (**b**) Images were quantified for different morphological parameters, including average mitochondrial number, average length, footprint, number of branched networks, and the number of branches per network. Each data point represents a cell color-coded by experiment. Columns represent mean (N = 3 biological replicates, 15–20 technical replicates, * *p* < 0.05 compared with control).

**Figure 5 cells-13-00631-f005:**
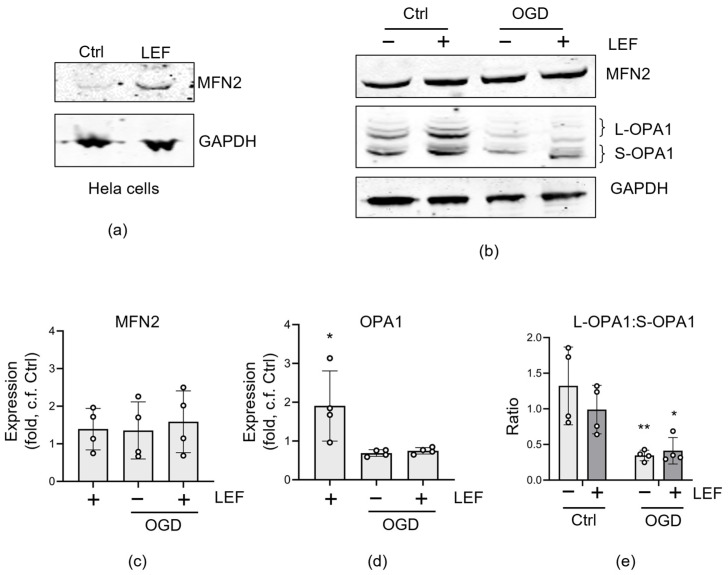
Leflunomide has limited effects on protein expression in C17.2 cells. HeLa (**a**) or C17.2 (**b**–**e**) cells were treated with Leflunomide (50 μM) for 16 h. (**a**) HeLa cell lysates were analyzed by Western blot for the expression of MFN2. (**b**) Lysates from C17.2 cells ± Leflunomide ± OGD were analyzed for the expression of OPA1 and MFN2. (**c**) MFN2 protein expression was not altered. (**d**) OPA1 expression increased in control cells (* *p* < 0.05, one-way ANOVA) after Leflunomide treatment, but this was not maintained following OGD. (**e**) the ratio of long(L)-OPA1: Short(S)-OPA1 was determined, and although there was a decrease by OGD (two-way ANOVA, * *p* < 0.05, ** *p* < 0.01), this was not reversed by Leflunomide treatment (mean ± SD, N = 4).

**Figure 6 cells-13-00631-f006:**
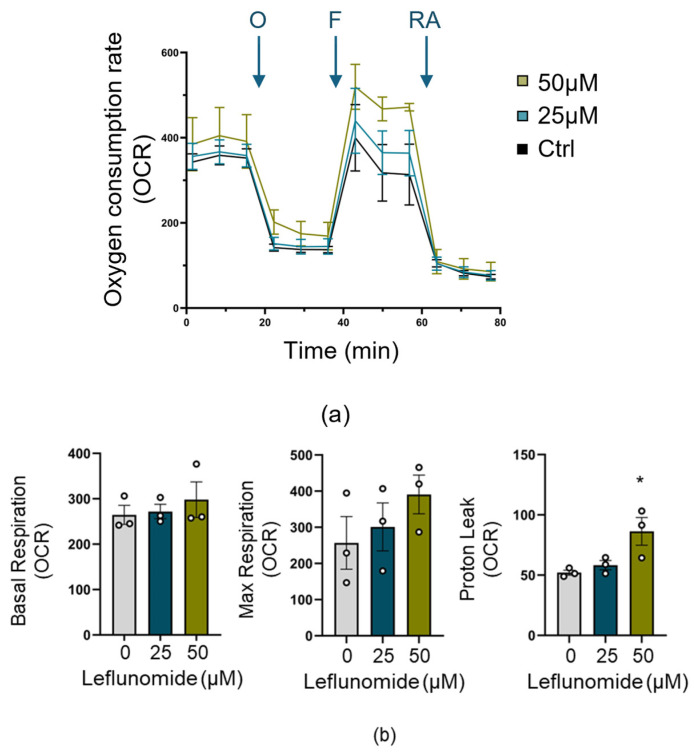
Leflunomide provokes proton leak in C17.2 cells. C17.2 cells were treated with Leflunomide for 16 h and then subjected to a Seahorse mitochondrial stress test to measure oxygen consumption. (**a**) Representative traces of treated C17.2 cells in a Seahorse experiment showing mean responses to oligomycin (O), FCCP (F), and rotenone/antimycin A (RA). (**b**) Graphs of calculated basal respiration, maximal respiration, and proton leak expressed relative to cell number with a scale factor of 200 (N = 3, * *p* < 0.05, one-way ANOVA).

**Table 1 cells-13-00631-t001:** Mitochondrial stress test assay.

Compound	Final Conc	Target
Oligomycin	1.5	ATP synthase
FCCP	1.0	Uncoupler
Rotenone/ Antimycin A	0.5	Complex I/ Complex III

## Data Availability

The data presented in this study are available on request from the corresponding author.

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
