# Peer review of "Leflunomide Treatment Does Not Protect Neural Cells following Oxygen-Glucose Deprivation (OGD) In Vitro"

_cells, 2024, doi:10.3390/cells13070631_

Round 1

Reviewer 1 Report

Comments and Suggestions for Authors

The study investigated the protective effect of leflunomide against nerve injury by glucose and oxygen deprivation. The researchers found that leflunomide did not promote the survival and regeneration of injured nerves. The overall research design is reasonable, the research method is correct, the correct results are obtained, and the mechanism of action is discussed. Although the study did not obtain positive results, the research on hypoxic-ischemic brain damage provides more evidence.

Reviewer 2 Report

Comments and Suggestions for Authors

Authors describe the research in preventing damage of hypoxia-ischemia using leflunomide which has been proved to have effects in OPA1 expression in an in vitro model with C17.2 cells. The findings for leflunomide in different cell lines are important to report, although the focus for preventing HI damage is not enough for the publication; title and discussion might be change to achieve this communication.

Figure 1 It would be better to have the arrows to the left removed. Post-OGD treatment - it is missing the reox timing

Figure 2 and 3. Standard deviation for control group is high. Figure 4: it is not clear if it is pre- or post-OGD treatment

Comments on the Quality of English Language

Revise line 218; all over the manuscript some words could be changed to improve the quality of the text.

Reviewer 3 Report

Comments and Suggestions for Authors

The manuscript by Curel et al. aimed to study the role of leflunomide in protecting the neural cells following oxygen glucose deprivation (OGD). These fundamental findings will greatly improve our understanding of the potential benefits of leflunomide if combined with additional research in the future.

Here lists a few of my concerns that could improve the overall quality of the manuscript if addressed appropriately.

Q1: Lack of novelty: The leflunomide incubation in C17.2 cells barely produced novel findings and promising results, even it’s not improving the OGD model, the authors failed to explore the exact mechanism of action of the leflunomide in these cells.

Q2: Justification for using the appropriate study model: C17.2 is a cancer cell line and needs strong clarification for being used as a good OGD study model to test the leflunomide. In additional to using immortalized cell line, I would suggest the authors to test leflunomide in primary neurons. The results from section 3.4 (Line 192-201) indicates the C17.2 cell line is not suitable for this study.

Round 2

Reviewer 3 Report

Comments and Suggestions for Authors

Thanks for your detailed response.